# Structures of the Varicella Zoster Virus Glycoprotein E and Epitope Mapping of Vaccine-Elicited Antibodies

**DOI:** 10.3390/vaccines12101111

**Published:** 2024-09-27

**Authors:** Wayne D. Harshbarger, Genevieve Holzapfel, Nishat Seraj, Sai Tian, Chelsy Chesterman, Zongming Fu, Yan Pan, Claire Harelson, Dongjun Peng, Ying Huang, Sumana Chandramouli, Enrico Malito, Matthew James Bottomley, James Williams

**Affiliations:** 1GSK, Rockville, MD 20850, USAnatasha.seraj@astrazeneca.com (N.S.); dpeng@its.jnj.com (D.P.); ying.huang@wuxibiologics.com (Y.H.); enrico.malito@schrodinger.com (E.M.);; 2WuXi Biologics, Cranbury, NJ 08512, USA; 3Moderna Therapeutics Inc., Cambridge, MA 02142, USA; 4Schrödinger, Inc., New York City, NY 10036, USA; 5Dynavax Technologies Corporation, Emeryville, CA 94608, USA

**Keywords:** vaccines, herpes zoster, antibodies, pathogen–host interaction

## Abstract

**Background:** Varicella zoster virus (VZV) is the causative agent for chickenpox and herpes zoster (HZ, shingles). HZ is a debilitating disease affecting elderly and immunocompromised populations. Glycoprotein E (gE) is indispensable for viral replication and cell-to-cell spread and is the primary target for anti-VZV antibodies. Importantly, gE is the sole antigen in Shingrix, a highly efficacious, AS01_B_-adjuvanted vaccine approved in multiple countries for the prevention of HZ, yet the three-dimensional (3D) structure of gE remains elusive. **Objectives**: We sought to determine the structure of VZV gE and to understand in detail its interactions with neutralizing antibodies. **Methods**: We used X-ray crystallography and cryo-electron microscopy to elucidate structures of gE bound by recombinant Fabs of antibodies previously elicited through vaccination with Zostavax, a live, attenuated vaccine. **Results**: The 3D structures resolve distinct central and C-terminal antigenic domains, presenting an array of diverse conformational epitopes. The central domain has two beta-sheets and two alpha helices, including an IgG-like fold. The C-terminal domain exhibits 3 beta-sheets and an Ig-like fold and high structural similarity to HSV1 gE. **Conclusions**: gE from VZV-infected cells elicits a human antibody response with a preference for the gI binding domain of gE. These results yield insights to VZV gE structure and immunogenicity, provide a framework for future studies, and may guide the design of additional herpesvirus vaccine antigens. **Teaser:** Structures of varicella zoster virus glycoprotein E reveal distinct antigenic domains and define epitopes for vaccine-elicited human antibodies.

## 1. Introduction 

Varicella zoster virus (VZV) is a lymphotropic and neurotropic alpha herpesvirus that infects humans and is the causative agent for chickenpox, and herpes zoster (HZ) also known as shingles. VZV is a member of the human Herpesviridae family, which includes herpes simplex virus 1 and 2 (HSV-1, HSV-2). HZ is a common and often debilitating disease that occurs primarily in older or immunocompromised individuals due to the reactivation of dormant VZV, which establishes life-long latent infection in the dorsal root and cranial ganglia after primary infection. Approximately one in three persons will develop HZ, and due to the onset of immunosenescence with increasing age, the risk for HZ also increases, and by the age of 85, more than 50% of people will be affected [1,2]. Acute symptoms include a painful rash and blisters, fever, light sensitivity, and itching. However, long-term complications can include post-herpetic neuralgia (PHN), which manifests as intense nerve pain and lasts long after resolution of the acute symptoms, causing physical disability, emotional distress, and can prevent normal day-to-day activities [3,4]. 

In 2006, zoster vaccine live (ZVL [Zostavax; Merck]), a live attenuated VZV vaccine, became the first vaccine to be approved for the prevention of HZ and PHN, in adults aged 50 years and older [5]. In 2017, the recombinant zoster vaccine (RZV [Shingrix; GSK]), a subunit vaccine that utilizes glycoprotein E (gE) and the adjuvant AS01_B_, was approved in the United States for the same age group and indication as ZVL. The very high efficacy of RZV, even in adults over 70, compared to the lower efficacy of ZVL led to the latter being discontinued in the US [6]. Approval of Shingrix has followed in multiple countries including Canada, the UK, China, Japan, Hong Kong, Australia, New Zealand, Singapore, and the countries of the EU. The indication for Shingrix has recently been expanded in the European Economic Area to include adults aged 18 or older who are at increased risk of shingles and was expanded in the US for the same age group for individuals who are or will be at increased risk for HZ due to immunodeficiency or immunosuppression caused by known disease or therapy. Phase III efficacy trials for Shingrix demonstrated age-independent protection against HZ and protective immunity has been found to persist for at least a decade post the two-immunization schedule [7,8,9,10,11]. 

While full-length VZV gE (referred to as gE_FL) is the key antigenic component of RZV, its structure and function are poorly understood. VZV gE is a type I membrane protein, is the most abundant viral glycoprotein expressed on the surface of VZV-infected cells, and has also been shown to be the primary target for antibodies (Abs) elicited after ZVL vaccination [12,13]. Based on amino acid sequence, the C-terminal region of gE appears to be conserved among alpha-herpes viruses. For herpes simplex virus (HSV), the C-terminal region has been shown to function as an Fc receptor for IgG (referred to in this paper as the Fc binding domain or gE_FcBD), thus potentially serving as an immune evasion mechanism by interfering with antibody-mediated viral clearance [14,15]. Unique to VZV gE is an N-terminal region (amino acids 1–188), which makes gE essential for viral replication, cell-to-cell spread, and secondary envelopment [16,17,18,19,20]. This N-terminal region has also been implicated in the binding to insulin-degrading enzyme (IDE), which may serve as a cellular receptor for the virus [21]. 

Critical for many VZV gE functions, as well as for VZV neurovirulence, is the formation of a non-covalent heterodimer with glycoprotein I (gI), an interaction that is conserved among gE homologues [22]. The gI binding domain (gIBD) on gE has been mapped to residues 178 through 206. Though not essential for replication, gI has been shown to be involved in post-translational modification and trafficking of gE, as well as being important for the gE immune evasion function via the proposed Fc-receptor activity and a bi-polar bridging mechanism [23]. 

It is thought that the main mechanistic driver of protection against HZ is gE-specific cell-mediated immunity (CMI), though no correlate of protection for Shingrix has been established [24]. The superiority of RZV over ZVL is thought to be due to the higher and more durable gE-specific memory Th1-type responses elicited by RZV [25]. Notably, several studies have demonstrated high durability of RZV-elicited immune responses and vaccine efficacy. A recent report showed that 10 years after receiving the two-dose RZV vaccine, gE-specific CMI responses and anti-gE antibody concentrations remained approximately 3.5-fold and 6-fold higher than pre-vaccination levels, respectively [11,26]. ZVL and RZV have been shown to elicit distinct Ab binding patterns, potentially due to either the presence or absence of O-linked glycans on gE [27]. This indicates that, at least to some extent, there is a unique Ab landscape after immunization with either of these vaccines. Interestingly, in children aged 12–22 months who were immunized with a live-attenuated VZV vaccine, anti-gE Ab titers were shown to be a correlate of vaccine-mediated protection against varicella [28]. Thus, it is possible that there is an underappreciated role that Abs play in protection afforded by either ZVL or RZV. 

The purpose of this study was to understand the molecular structure of VZV gE and to map epitopes targeted by anti-gE human Abs that were elicited through ZVL vaccination. We provide a comprehensive structural characterization of VZV gE, comprising a total of three X-ray structures, a low resolution cryo-EM map, plus epitope binning experiments, thereby uncovering five antigenic sites spread across the gIBD and the FcBD of gE. These data suggest the gIBD to be the primary target for anti-gE Abs elicited through ZVL vaccination and reveal a bias for Abs recognizing one face of the gE protein. Altogether, this work delivers an antigenic map and structurally defined epitopes on VZV gE, describes gE reagents that can be used to further interrogate the gE Ab responses from either vaccination or natural infection, and provides a blueprint for epitope mapping that could be extended to other herpesviruses. 

## 2. Methods

### 2.1. Expression and Purification of VZV gE_FL and gE_sub 

Plasmid for FL gE or gE_sub, both of which include a C-terminal 6xHis-tag, was transfected into and expressed by Expi293 cells (Catalog # A14635, Gibco, Miami, FL, USA). The harvested cell culture supernatant of was loaded onto a 5 mL HisTrap HP column (Catalog # 17524801, Cytiva Life Sciences, Marlborough, MA, USA). After loading, the column was washed with buffer A (50 mM Tris pH 8.0, 300 mM NaCl, 20 mM Imidazole) for 20 column volume (CV) to remove non-specifically bound proteins. Elution was carried out with buffer gradient from 0% to 50% buffer B (50 mM Tris pH 8.0, 300 mM NaCl, 500 mM imidazole) within 10 CVs. The fractions containing the protein of interest were pooled and concentrated. Sample was further purified over a pre-equilibrated HiLoad 16/600 Superdex 200 pg column (Catalog # 28989335, Cytiva Life Sciences) with 25 mM Tris, pH7.5, 150 mM NaCl as running buffer. The fractions corresponding to either gE FL or gE_sub, based on SDS-PAGE analysis, were pooled and quantitated using the absorbance at 280 nm.

### 2.2. Expression and Purification of Fabs

Anti-gE Fab variable regions were cloned into vector backbones containing the human IgG1 Fab constant region. A TEV cleavable Strep Tag II sequence was appended to the C-terminus of the heavy chain. Fabs were transiently expressed in HEK293 Expi cells and the supernatants were diafiltrated to remove biotin before being loaded onto a StrepTrap HP column (GE Healthcare, Chicago, IL, USA), followed by size-exclusion chromatography.

### 2.3. Limited Proteolysis of gE 

Limited proteolysis of FL gE was done using the Proti-Ace Kit (Catalog # HR2-429-01, Hampton Research, Aliso Viejo, CA, USA). The assay was performed following the vendor instructions. VZV gE was mixed with the individual proteases at a mass ratio of 1000:1 and incubated at 37 °C for 60 min. Protease-cleaved fragments were characterized by SDS-PAGE gel. Protein bands for fragments of interest were transferred to PVDF membrane and analyzed by liquid chromatography with tandem mass spectrometry to identify possible cleavage sites. 

### 2.4. Liquid Chromatography with Tandem Mass Spectrometry (LC–MS-MS)

The excised gel bands were cut into cubes (ca. 1 × 1 mm) and processed following the in-gel digestion protocol [29]. The gel pieces were reduced with DTT and alkylated with iodoacetamide. Promega trypsin was used to digest the processed gel bands overnight at 37 °C. The extracted peptides were dried down using SpeedVac and reconstituted in mass spectrometry grade water with 0.1% (*vol*/*vol*) formic acid for reverse-phase liquid chromatography with tandem mass spectrometry (RPLC–MS/MS) analysis. An Orbitrap Fusion mass spectrometer (ThermoFisher Scientific, Waltham, MA USA) coupled to Vanquish UHPLC system was used. UHPLC was set at flow rate of 0.25 mL/mL with a gradient of 2–40% of acetonitrile from 0–45 min. The mass spectrometer was operated in a data-dependent mode with full-scan MS spectra acquired in the Orbitrap analyzer at a 120,000× resolution, followed by dynamic ddMS^2^ acquisition in the ion trap. The raw data were used to against VZV gE sequence using Protein Matrix Software (PMI V4.4). The search settings included trypsin as the digestion enzyme, parent ion tolerance of 10 PPM, and fragment ion tolerance of 0.6 Da, carbamidomethyl of cysteine as a fixed modification, and oxidation of methionine as a variable modification. 

### 2.5. Epitope Binning by HPLC

Epitope binning was performed using HPLC–SEC. VZV gE was mixed with primary antibody Fabs at mass ratio of 1:3 for 10 min at room temperature to make gE-primary Fab complexes. These mixed samples were further mixed with secondary Fab at the same ratio to make gE-primary Fab-secondary Fab complex. These complexed samples were injected by Waters Alliance HPLC and eluted through Superdex 200 Increase 3.2/300 SEC column (Catalog # 28990946, Cytiva Life Sciences). The formation of gE-primary Fab-secondary Fab complex would indicate that these two Fabs do not compete for binding to the same epitope.

### 2.6. Biolayer Interferometry and K_D_ Determinations 

Binding kinetics of gE antibodies to gE was determined by Octet Red96 (Sartorius) BLI using 1 × PBS with 1% BSA as binding buffer. Fabs were loaded onto FAB2G biosensors (Catalog # 18-5125, Sartorius) to reach capture level between 0.5 to 1.0 nm. Fab-coated biosensors were further dipped into gE analytes at concentration ranging from 25 nM to 1.6 nM to generate binding curves. The binding curves were generated and reported by GraphPadPrism software using raw data exported from Octet Data Analysis software version 11.0. 

### 2.7. Crystallization and Structure Determination of gE_FcBD:Fab 5A2 Complexes 

Purified gE_FcBD was incubated with a 1:2 molar excess of Fab 5A2 overnight at 4 °C. Complex of gE_FcBD:5A2 was separated from excess 5A2 Fab via size-exclusion chromatography in running buffer containing 10 mM Hepes pH 7.5, 150 mM NaCl, and 5% glycerol. Protein was concentrated to 5 mg/mL and crystals of the gE216:5A2 Fab complex were produced by hanging drop vapor diffusion at a protein to buffer to additive ratio of 1:1:0.2 with buffer containing either 2.0 M ammonium sulfate, 0.1 M Bis-Tris pH 6.5, and additive benzamidine hydrochloride at 100 mM stock concentration for the non-cleaved structure, or buffer containing 0.1M Bis-Tris, pH 6.5, and 25% *w*/*v* polyethylene glycol 3350 for the cleaved structure. In either case, crystals were harvested, cryoprotected in their respective mother liquor that was supplemented with 20% glycerol, flash frozen, and shipped for data collection on the SER-CAT 22-ID beamline at the Advanced Photon Source. 

X-ray data were processed using HKL3000 [30] and in each case found to contain high mosaicity. The structures were determined by molecular replacement (MR) with Phaser [31]. Homology models for the gE_FcBD and Fab 5A2, generated from Swiss-Model and the PIGS server [32], respectively, were used as MR search models. Iterative rounds of refinement and model building were achieved using Phenix [33] and Coot [34]. Graphics were created using ChimeraX [35] and Pymol [36]. Refinement statistics for each structure are in Appendix A.

### 2.8. Expression of gE_gIBD:Fab Complexes

Purified gE_gIBD was co-expressed with Fab in Expi293F GnTI- HEK (Gibco catalog # A39240) mammalian cells by transient co-transfection. Expi293F GnTI- HEK cells, derived from Expi293F cells but engineered to lack N-acetylglucosaminyl transferase for uniform glycosylation patterns, were maintained in Expi293 Expression media (Gibco catalog # A14351-01) and grown at 37C, 8% CO_2_, 80% humidity. The day before transfection, cells were seeded at 2.4 × 10^6^ cells/mL after a complete media exchange. On the day of transfection, the cell density and viability were measured by trypan blue exclusion on a ViCell XR cell viability analyzer. The culture was adjusted to 3.6–3.8 × 10^6^ cells/mL by medium feed or culture split. 

DNA plasmids for transfection were diluted in prewarmed Opti-MEM media (Gibco catalog # 11058-021) in a volume that was 5% of the transfection volume. For the complex, 50% of the mass of DNA transfected was gE_gIBD expression plasmid. Each antibody, comprising of a separate heavy chain and light chain DNA plasmid¸ made up 25% of the mass transfected. For the final DNA concentration in Opti-MEM, a microgram of DNA was co-transfected for every milliliter of transfection culture. The diluted DNA was filtered through a 0.2 µm filter. ExpiFectamine 293 reagent (from Gibco ExpiFectamine 293 Transfection Kit, 1L, catalog # A14524) was diluted into a volume of prewarmed Opti-MEM media corresponding to 5% of transfection volume of prewarmed Opti-MEM. 

For a 1000 mL transfection culture, 500 µg of gE_gIBD expression vector, 125 µg of Fab heavy chain, 125 µg of Fab light chain expression vector combined in 50 mL Opti-MEM. In another 50 mL of Opti-MEM, 2.7 mL of ExpiFectamine was diluted. Five minutes after the dilution of ExpiFectamine reagent in Opti-MEM, the diluted DNA solution was added to the diluted ExpiFectamine to form the DNA ExpiFectamine complex for transfection. Twenty minutes after the addition of DNA to the ExpiFectamine reagent, the complex was added to the cell culture. Sixteen to twenty-two hours after the addition of diluted DNA to the cells, Enhancer 1 and 2 from the same kit were added to cell culture. Four to six days post transfection, cell count, and cell viability were measured with ViCell XR cell Viability Analyzer. When cell viability is below 80%, cell media supernatants were harvested. A 1L transfection culture was divided into 3 × 500 mL Corning centrifuge tubes (Corning catalog 431123) and placed in Beckman Avanti J HC floor centrifuge with JS4.2 rotor in alternate buckets. The culture was centrifuged at 4200 rpm for 50 min at 10 °C and the supernatant was poured off into a clean 1L storage bottle. The supernatant was weighed to determine harvest volume and stored at 4 °C or −20 °C until purification.

### 2.9. Purification of gE_gIBD:Fab Complexes

The culture supernatant for Fabs co-expressed with gE_gIBD were loaded onto a nickel–nitrilotriacetic acid Excel column (Cytiva Life Sciences). The column was washed with buffer containing 350 mM sodium chloride and 20 mM TRIS (pH 7.5) prior to elution. Protein was eluted with buffer composed of 150 mM sodium chloride, 500 mM imidazole, and 20 mM TRIS (pH 8.0). Protein eluted at 25 mM imidazole. Fractions containing gE_gIBD:Fab complex were then combined and concentrated prior to further purification by size-exclusion chromatography (SEC). SEC was performed using a Superdex 200 Increase 10/300 GL (Cytiva Life Sciences) with 150 mM sodium chloride, 10 mM Hepes (pH 7.5). Further purification was performed by SEC using a Superdex 75 Increase 10/300 column (Cytiva Life Sciences). Fractions containing complex were separated from free Fab and protein aggregates and the complex was confirmed by SDS-polyacrylamide gel electrophoresis analysis. Fractions containing gE_gIBD:Fab complexes were pooled and concentration was determined using the absorbance at 280 nm. 

### 2.10. Crystallization and Structure Determination of gE_gIBD:1E3 Fab and gE_gIBD:1A2:1E12 Fab Complexes 

Crystals of the gE_gIBD:1E3 Fab complex were obtained via sitting-drop vapor diffusion. Protein was mixed at a 1:1 ratio of protein (8 mg/mL) to reservoir solution (1.9 M ammonium sulfate, 0.1 M citric acid pH 5.6). Crystals appeared prior to 14 days. Cryo-protectant (well solution supplemented with 20% *v*/*v* PEG400) was added to the drop prior to flash freezing in liquid nitrogen and shipped for data collection. Data were collected at National Synchrotron Light Source II (NSLS-II) at Brookhaven National Laboratory. Protein crystals diffracted X-rays and were processed and scaled to a maximum resolution of 1.90 Å in space-group C2221 using AutoProc [37]. The structure was solved by molecular replacement with Phaser [31]. A Fab 1E3 homology model and portions of a homology model for the gE_gIBD were used as search models. The 1E3 CDRs and the remainder of unmodelled gE_gIBD were built manually in Coot [34]. A single copy of gE_gIBD:1E3 Fab was present in the asymmetric unit. Iterative rounds of reciprocal and real space refinement were performed using PHENIX Refine [33] and Coot [34]. 

Crystals of gE_gIBD:1A2:1E12 ternary complex were obtained using the sitting drop vapor diffusion method using a 1:1 ratio of protein to reservoir solution at a protein concentration of 10 mg/mL. Crystals appeared after 45 days. The highest diffracting crystal grew in 25% PEG 3350 and 0.1 M lithium sulfate and was cryoprotected in mother liquor supplemented with a final *v*/*v* concentration of 20% ethylene glycol. Data were collected at the Advanced Photon Synchrotron (APS) at Argonne National Laboratory, beamline 22-ID-E. X-ray data were processed and scaled into space-group C2 to a maximum resolution of 3.09 Å using HKL3000 [30]. The structure was solved using Phaser [31] with the gE_gIBD from the Fab 1E3-bound structure as one search model, and homology models of Fabs 1E13 and 1A2 as additional search models. One gE_gIBD molecule bound by one Fab 1E12 and one Fab 1A2 was observed in the asymmetric unit. Iterative rounds of refinement were performed using PHENIX Refine [33] and Coot [34]. The final structures for both complexes had acceptable R-factors, geometry, and Ramachandran statistics (Appendix A).

### 2.11. Cryo-EM of VZV gE_FL:5B3 Fab 

A 3.0 µL aliquot of purified VZV gE FL:5B3 Fab complex at a concentration of 0.4 mg/mL was applied to freshly glow-discharged Quantifoil 1.2/1.3–400 mesh copper grids, blotted, and plunge frozen into liquid ethane using an FEI Vitrobot Mark IV vitrification apparatus (ThermoFisher Scientific, Waltham, MA USA). Vitrified grids were loaded into a Titan Krios operating at 300 keV and equipped with a Gatan K3 direct electron detector. A total of 3748 movies were collected using a pixel size of 0.83 Å/pix and total dose of 49.28 e^−^/Å^2^. Movies were imported into the Computational Imaging System for Transmission Electron Microscopy (cisTEM) software version 1.0 for single particle analysis. Briefly, movies were aligned using the Unblur algorithm followed by CTF determination using CTFFIND [38,39]. A total of 33,015 particles were selected from the micrographs and were subjected to 2D classification, which resulted in the isolation of ~17,000 particles that resembled the expected morphology of 5B3 bound to gE. The isolated particles were next subjected to asymmetric 3D auto-refinement, resulting in a 15 Å resolution map according to gold-standard FSC at 0.143 criteria.

## 3. Results 

### 3.1. Stability of VZV gE, the gEgI Heterodimer, and Engineered gE Fragments 

Extensive attempts to crystallize the full-length gE ectodomain (gE_FL), which includes a predicted disordered N-terminal region, were unsuccessful. To assist with structural biology efforts, we sought to identify gE fragments that could be readily recombinantly expressed and purified for structural and functional studies. First, we used limited proteolysis to remove flexible regions of gE_FL. We screened a panel of proteases and found that digestion with subtilisin resulted in a gE molecular weight (MW) band on SDS-PAGE that was ~25 kDa smaller than gE_FL (Appendix A). Extraction of the protein from the gel, followed by LC–MS/MS analysis, indicated that the subtilisin cleavage site was between residues 115–116, thus removing a large part of the N-terminal region (Figure 1A and Appendix A). The region from residue 116 onwards was generated as a new expression construct, named gE_sub, and was readily purified as a single, soluble peak via size-exclusion chromatography (SEC) (Appendix A). Next, we performed homology modeling of gE and found that only residues 301–516, which encompass the C-terminal portion of gE and is annotated as an Fc-binding domain (FcBD), were able to generate a predicted structure. This structure was based on the ~30% sequence identity with, and X-ray structure of, HSV-1 gE. A corresponding expression construct, termed gE_FcBD, was also produced and found to purify as a single peak via SEC (Appendix A). 

A higher melting-point (Tm) can increase the probability of crystallization. Here, we used differential scanning fluorimetry (DSF) to assess the thermostability of these constructs. For comparison, we also tested gE_FL that was either fully glycosylated (gE_FL+) or did not contain complex N-linked glycans (gE_FL−) at positions Asn266 and Asn437, as well as with gE in complex with gI (gEgI). For the gEgI heterodimer, the complex was verified to be present via SEC followed by SDS-PAGE analysis (Appendix A). Notably, gE_sub melted at the same temperature as gE_FL+ and gE_FL− (Tm 53 °C), supporting the prediction that the N-terminal region is highly flexible and, therefore, does not contribute to overall thermostability (Figure 1B). Interestingly, gE_FcBD and the gEgI heterodimer were found to have Tm values 14 °C and 13 °C, respectively, higher than gE_FL. Taken together, this suggests that the gE domain that binds gI (termed the gIBD) is inherently less stable than the FcBD (Figure 1B). Curiously, there was no distinct thermal transition attributable to gI, which suggests that this glycoprotein is either highly flexible/disordered or that there are no buried aromatic residues, which is a requirement for the thermostability assay used, or that gI is intricately complexed with gE and, therefore, does not give rise to a unique signal separable from the peak at 66 °C. 

Despite the modifications to gE, including deglycosylation or the removal of flexible regions and/or regions with lower thermostability, attempts to crystallize either gE_FL, gE_sub, gE_FcBD, or the gEgI heterodimer were unsuccessful. Therefore, we hypothesized that stabilization of gE, in particular the gIBD, through complex formation with an antibody (Ab) fragment antigen-binding domain (Fab), may be beneficial and sought to generate such Ab reagents. 

### 3.2. Binding Affinity and Epitope Competition of ZVL-Elicited Abs 

We were unaware of any antibody sequences having been determined for anti-gE antibodies elicited after vaccination with RZV. However, sequences for several anti-gE antibodies that were elicited after vaccination with ZVL were reported by Sullivan et al. and, thus, were selected as tools for our structural studies [13]. We randomly selected four of these anti-gE human antibodies (named 1A2, 5A2, 1E3, and 5B3) for which the corresponding recombinant IgGs had been previously characterized in a VZV neutralization assay and shown to neutralize only in the presence of complement [13]. We produced and purified the recombinant Fabs for these Abs and determined that each bound to gE_FL with high affinity, with equilibrium dissociation constant (K_D_) values in the low nanomolar range (Appendix A and Appendix A). Additionally, we tested the Fabs for cross-site competition and found that 1A2 and 5A2 bound different epitopes, whereas 1E3 and 5B3 competed for binding to gE (Figure 1C). Interestingly, all four Abs bound to gE_sub, while only 5A2 bound to gE_FcBD, thus identifying Fabs that might potentially stabilize, and chaperone crystallization of, different gE regions (Figure 1C). 

### 3.3. Structure of gE_FcBD in Complex with Fab 5A2

Initial attempts to crystallize the gE_FcBD:5A2 Fab complex resulted in a protein crystal with high mosaicity that diffracted X-rays to a maximum resolution of 4.3 Å. This structure was solved by molecular replacement using the gE_FcBD homology model and a model of the 5A2 Fab as search template models. Visible electron density extended from the gE_FcBD N-terminal residue Gln305 through to residue Phe479 and covered the majority of Fab 5A2 (Figure 2A–F and Appendix A). Despite the relatively low resolution, the electron density was sufficient for placing main chain and side chain residues, particularly at the antibody–antigen interface (Appendix A). The structured C-terminal domain of gE was virtually structurally identical to the reported X-ray structure for HSV1-gE (PDB 2GIY), with a room mean square deviation of 1.9 Å across 175 C_α_ atoms (Figure 2G). The overall fold for this domain consists of thirteen beta-strands (named here as β1′–β13′) organized into three beta-sheets, two of which pack in the center and form an Ig variable-like fold (Figure 2A,B). Three naturally occurring disulfide bonds, which are also conserved in HSV1 gE, are located between residues C357:C383, C366:C375, and C402:C412. Overall, the high conservation of the FcBD between VZV and HSV-1 underscores the potentially shared function. 

Strands β6′ and β13′ are not part of the Ig variable-like fold; rather, they are located on what appears to be a flexible extension away from the more tightly packed Ig-variable-like core of gE_FcBD and consist of several loops and two single turn α-helices (α1′–α2′) (Figure 2B). Interestingly, a second data set that we were able to obtain for the gE_FcBD:5A2 complex resulted in a 3.5 Å resolution structure and we found that β6′ and β13′ were completely missing from the electron density maps. Specifically, residues Asp354 through Ser387, and residues after Thr466 had no electron density (Figure 2B). Though we initially thought this may be due to flexibility, examination of the crystal packing indicated that there was not sufficient space between molecules in the unit cell to accommodate this portion of gE. It is possible that residual contaminating proteases that were not removed during purification cleaved gE during crystallization. Serendipitously, the cleavage event permitted growth of higher quality crystals that allowed for a more thorough analysis of the gE–Fab interface, as the side chain densities were much clearer (Appendix A and Appendix A). 

### 3.4. Molecular Details for Fab 5A2 Binding to gE_FcBD

Fab 5A2 recognizes a large, conformational epitope, burying ~1000 Å^2^ on the surface of gE, mainly localized across beta sheets β1′–β4′ (Figure 2C). All three heavy-chain (HC) complementarity-determining regions (CDRs) are involved in binding, as are light-chain (LC) CDRs L1 and L3. The epitope includes a cluster of histidine residues made up of His312, His317, His319, and His332, which interact with several arginine and tyrosine side chains across each of the 5A2 HC CDRs (Figure 2D). The electrostatic surface of the epitope is predominantly negative, which is complemented well by the many polar and charged residues on the 5A2 paratope (Figure 2E). The 5A2 residue Arg103_HCDR3_ wedges into a pocket enclosed by His332, Glu334, His312, and Trp314. Interestingly, His312 and His332 are structurally located near His247 in HSV1 gE, which has been suggested to play a role in mediating pH-dependent binding to Fc [23] (Appendix A). Though binding of VZV gE to Fc has not been experimentally reported, we structurally aligned our gE_FcBD:5A2 Fab structure with the model of the FcBD for HSV1 gE in complex with an antibody Fc (PDB 2GJ7). This alignment shows that the VZV gE residue His332 may recognize the same pocket at the interface with Fc, as the HSV1 gE His247 is predicted to recognize (Appendix A). 

There are a total of fourteen predicted hydrogen bonding interactions between gE and 5A2, nine of which can be attributed to the 5A2 HC (Figure 2F). Notably, the His312 side chain forms π–π stacking and a hydrogen bond with the side chain from LC residue Tyr32_LCDR1_, whereas His332 forms a hydrogen bond with the side chain of Tyr52_HCDR2_, and His317 hydrogen bonds with the sidechain for Asn58_HCDR2_. Additional hydrogen bonds from the HC come from the side chains for residues Thr56, Thr57, and Arg103, as well as main chain oxygen or nitrogen for residues Thr57, Gly97, and Glu102 (Figure 2F). For the LC, Tyr32 can form a second hydrogen bond with the side chain for Gln334. The side chain for Ser93_LCDR3_ forms a hydrogen bond with the side chain for Asn315 whereas the remainder of the LC hydrogen bonds are attributed to the main chain oxygen or nitrogen for Ser91, Tyr92, and Ile94 (Figure 2F). Interestingly, despite the high structural conservation with HSV1-gE discussed above, the loop between strands β1′ and β2′ is perturbed in the alignment between these two proteins, forming a “U” from residues His312 to His317. Without an available unbound VZV gE_FcBD structure, it is not possible to know for sure, but the distortion of gE in the structures does suggest some plasticity in the 5A2 epitope. 

### 3.5. Structure of the gE–gI Binding Domain in Complex with Fab 1E3

Attempts to crystallize gE_sub bound by one of our Fabs were unsuccessful; therefore, armed with the gE_FcBD structure, we designed a new gE construct that initiated at the subtilisin cleavage site and terminated at the first visible residue in the N-terminus of our gE_FcBD structure (Gln305). Expressed alone, this construct (which we called gE_gIBD) did not form soluble protein. However, when co-expressed with either Fab 1A2, 1E3, or 5B3, we were able to recover gE_gIBD:Fab complexes (Appendix A). Crystals of the complex between gE_gIBD:1E3 Fab diffracted X-rays to a maximum resolution of 1.9 Å. The structure was solved by molecular replacement using a Fab 1E3 homology model and portions of a homology model for the gE_gIBD as search templates. The 1E3 CDRs and the remainder of unmodelled gE_gIBD, all of which had clear electron density for main chain and side chain residues, were built manually in Coot (Figure 3A–C and Appendix A). 

The structure of gE_gIBD reveals an IgG-like fold composed of nine β-strands (β1–β9) and two α-helices (α1-α2) (Figure 3B). The β-strands form two distinct β-sheets, which are folded on top of each other; β-sheet one includes β1, β3, β4 β5, β8, and β9, whereas β-sheet two is formed by β2, β6, and β7. The N- and C-terminal residues ranging from 116–139 and 295–305, respectively, are not visible in the electron density. Additionally, the stretch of residues from 177–211, which is the expected binding region for gI, is not visible, suggesting high flexibility. 

The structure reveals that Fab 1E3 recognizes the face of gE_gIBD that encompasses β2, β6, and β7, utilizing both HC and LC CDRs to bury ~1050 Å^2^ of surface area. The majority of the buried surface area (BSA) derives from the HC, where the antiparallel β-strand formed by the HC frame-work region 2 (HFR2)-H2-HFR3 region of 1E3 buries over 300 Å^2^ and binds perpendicular to the β-strands that form the epitope. The paratope/epitope interface is dominated by polar interactions, and several arginine residues from Fab 1E3, namely, HC residue Arg67 and LC residues Arg30 and Arg93, recognize large negative patches on the gE surface (Figure 3D). Tyrosine, phenylalanine, and a tryptophan residue on the Fab 1E3 HC all recognize pockets that are formed between β2–β7 or β6–β7 on gE. Though the 1E3 HC buries almost twice as much surface as the LC, the twenty hydrogen bonds between gE and 1E3 are split evenly between each, suggesting that both chains are critical for binding (Figure 3F–H). Most of the hydrogen bonds are with gE residues on the loops between β-strands, with the exception being a stretch of HC residues Asp58, Thr59, and Tyr 60, which form a total of five hydrogen bonds with β-2 and β-7 (Figure 3F). 

### 3.6. Structure of gE_gIBD in Ternary Complex with Fabs 1A2 and 1E12

Encouraged by these results, we sought to elucidate additional epitopes for the ZVL-elicited Abs; however, crystallization attempts with 1A2 and 5B3, were unsuccessful. Therefore, we produced the Fabs for two additional Abs, 1E12 and 1D7, and binned them against Fabs 1E3, 1A2, and 5A2 (Appendix A). Interestingly, this revealed that Fabs 1E12 and 1D7 competed with each other and with Fab 1E3, but not with 1A2 or 5A2. As we showed that 5B3 also competes with 1E3, this additional binning reveals that five out of the six Abs selected for this study recognize the gIBD, and that the majority of them cluster near the 1E3 binding site.

In attempts to further stabilize the gE_gIBD, we co-expressed ternary complexes with Fab 1A2 paired with either Fab 1E3, 1E12, or 1D7. This approach ultimately resulted in crystals of the gE_gIBD:1E12:1A2 ternary complex, which diffracted to a resolution of 3.1 Å (Appendix A). Molecular replacement using gE_gIBD from the 1E3-bound structure and homology models for either Fab yielded a structure of the ternary complex with clear electron density, especially for side chain residues at the epitope/paratope interface (Figure 4A and Appendix A). The loop connecting β4–β5, which is disordered in the Fab 1E3-bound structure (residues 221–228), had clear electron density, which allowed for building of these residues. However, the loop that interacts with gI, encompassing residues 177–213, was still highly flexible and, thus, not visible (Figure 4A). 

The structure reveals that Fab 1E12 recognizes a large, conformational epitope that buries over 800 Å^2^ on gE covering portions of β2 and β7, as well as the loop connecting β1–β2. This epitope mapping explains the strong binding competition with 1E3, and superposition of the 1E12- and 1E3-bound structures shows that the antibodies would clash with their respective HCs (Appendix A). Despite the strong binding competition, each Ab recognizes a unique epitope on gE. Unlike the 1E3 epitope, which is dominated by negatively charged patches, the surface of gE recognized by 1E12 is predominantly hydrophobic or positively charged (Appendix A). The 1E12 HC accounts for all eleven of the hydrogen bonds with gE_gIBD and utilizes all three CDRs, with CDR H3 and H2 combining for over 500 Å^2^ of the BSA. Similar to 1E3, the majority of the hydrogen bonds are found on the loops between β-strands, and 1E12 forms seven of its hydrogen bonds with the loop between β1–β2. The additional hydrogen bonds are formed between the CDR H3 and the main chain atoms for Thr153 and Arg155 on β2, and the CDR H2 with the main chain for Leu261 (Figure 4D). 

Also consistent with the binding competition assay, Fab 1A2 was revealed to bind an epitope that is distant from either the 1E3 or 1E12 binding site, burying ~600 Å^2^ across portions of β3, β4, and β9 (Figure 4E). Fab 1A2 binds gE_gIBD entirely through HC contacts, utilizing all three HC CDRs, with CDR H3 accounting for ~300 Å^2^ of BSA (Figure 4F). The epitope is predominantly non-charged, though two protruding arginine residues (Arg165 and Arg223), located at the periphery of the epitope, combine for four of the nine hydrogen bonds with gE, all with the CDR H3 residue Glu99 (Figure 4G). Additional hydrogen bonds are found with the side chains for CDR H2 residues Arg50 and Asn53 with the main chain oxygens for Asp279 and Gln159, respectively, as well as the main chain of CDR H1 residue Glu31 with the main chain nitrogen for Tyr162, and finally the main chain nitrogen for CDR H3 residue Glu99 with the mainchain oxygen for Gly163. 

### 3.7. Low-Resolution Cryo-EM Epitope Mapping of Fab 5B3 to Full-Length gE

Finally, we sought to visualize the approximate orientations of the C-terminal gE_FcBD and N-terminal gE_gIBD relative to each other in gE_FL. Screenings of cryo-EM grids of gE_FL in complex with various Fabs all reveal aggregated particles, with the exception of the gE_FL:5B3 Fab complex, for which we were able to generate 2D class averages and elucidate an ~15 Å resolution map (Figure 5A,B). Using the Fab binning data as a guide, we fitted the gE_gIBD and gE_FcBD X-ray structures, as well as a model for the 5B3 Fab, where a “donut-hole”, indicative for Fab density, was clearly visible even in the low-resolution EM map. Interestingly, though the gIBD was not visible in either of our gE_gIBD X-ray structures, the EM map indicated a blob of density that was not accounted for in our fitted model and extends out from residues Thr179 to Thr212, which agrees with where the gIBD should be located (Figure 5A). There was no EM density present for the disordered N-terminal region spanning residues 1–115, despite using gE_FL for the EM analysis.

Alignment of each gE domain in the gE_FL:5B3 Fab model with the solved gE_FcBD:5A2 and gE_gIBD:Fab X-ray structures, and visualization of gE_FL with all Fabs bound, provide further validation for how the individual gE domains were fitted, as the placement of the Fabs are in agreement with the binding competition results (Figure 5C). Specifically, the 5B3 and 1E12 epitopes each overlap/compete with 1E3, whereas neither 1A2 nor 5A2 clash with any of the other Abs. In summary, the cryo-EM data provide a glimpse of the probable structure for gE_FL, allowing for a holistic visualization of the epitope map for the anti-gE Abs described in this work. 

## 4. Discussion

The structure of VZV gE, the glycoprotein that is the primary target for Abs elicited through natural HZ infection, and the antigen in the AS01_B_-adjuvanted shingles vaccine RZV (Shingrix; GSK), proved challenging to resolve, presumably due to high flexibility. The lack of VZV gE structural information has prevented a detailed molecular description of immunogenic determinants, which may enable a better understanding of Ab-mediated protection against VZV. To circumvent these challenges, we first used computational modeling and enzymatic digestion to design truncated gE constructs that would be more prone to crystallization. As this alone was not sufficient, we further stabilized gE utilizing Fabs from anti-gE Abs that were previously isolated from ZVL vaccinees [13]. This approach, aside from successfully leading to the elucidation of the gE structure, also enabled a unique opportunity to define the 3D landscape on gE for vaccine-elicited human Abs. 

Comparing the structure of the VZV gE_FcBD with the previously determined structure for HSV-1 gE of the same domain, yielded the anticipated result; very high structural homology (Figure 1). Interestingly, we found that of the six anti-gE Abs that were selected for this study, only one, Ab 5A2, recognized the FcBD. One possibility for the apparent low frequency for FcBD Abs could be the nature of the ZVL vaccine. In a live attenuated vaccine, the gEgI heterodimer would likely be the predominant species on the cell surface and accessible to interact with B-cell receptors (BCRs), thus also enabling an active immune evasion function. As a result, BCRs recognizing the 5A2 epitope may have to compete with Ab Fc’s for binding. Indeed, we determined that the gE/gI heterodimer is more stable than gE alone, which may further favor generation of Ab responses to the heterodimer rather than to gE alone.

The epitopes for Abs 1E3, 1E12, 1A2, and 5B3 were all found to decorate the gIBD, and have angles of approach and epitopes that do not contact the loop that is used for forming a heterodimer with gI. This observation is further evidence for an intact gE/gI heterodimer as the main antigen in ZVL as the loop would be shielded from binding to Abs. The high resolution for the 1E3- and 1E12-bound structures informs molecular level details for overlapping epitope footprints and convergent interactions such as each Ab making hydrogen bonds with the side chain for His150. Informed by these data, future studies might use site-directed mutagenesis to delineate antigenic hot spots or how mutations of specific residues, such as His150, impact the antibody response. Additionally, it will be informative to compare the epitopes recognized by Abs elicited via ZVL vaccination with those from Shingrix vaccinees that received only the gE antigen. Together, these comparisons would facilitate a comprehensive understanding of the similarities or differences in the antigenic landscape for these vaccines. Our engineered gE_sub and gE_FcBD constructs will be useful tools to probe the magnitude and quality of the B-cell and T-cell responses to the various gE domains. The structures and Abs presented in this study can also support computational protein design efforts utilizing programs such as PROSS [40]. Such computational tools could be used to improve gE thermal stability and/or expression levels while ensuring that known epitopes or key residues remain intact. 

The antibodies used in this study were all shown previously to function through complement-dependent neutralization of VZV with IC50 values of less than 1 μg/mL [13]. Antibodies elicited through vaccination with either ZVL, or to an even greater degree with RZV, have also been shown to have antibody-dependent cellular cytotoxicity (ADCC) activity [41]. Alternatively, anti-gE complement-independent neutralizing Abs are not believed to have an impact on protection from VZV, as the virus readily spreads cell-to-cell, and even Abs found to recognize the N-terminal domain (responsible for cell-to-cell spread) have been shown to need complement for neutralization [42,43,44]. Based on this work, additional structural and epitope mapping studies, combined with Ab functional assessments, are currently ongoing, which will continue to provide insights to the humoral immune response to VZV gE. Further, these studies may guide the design of additional herpesvirus antigens against pathogens like HSV, for which effective vaccines do not yet exist. 

## 5. Conclusions

In this study, we developed multiple constructs of the varicella zoster virus glycoprotein E (gE), determined the binding affinities and binding competition profiles for a panel of human anti-gE ZVL vaccine-elicited antibodies, and solved the X-ray structures for Fabs from four of the Abs in complex with the gE constructs. These experiments provide the first 3D structural details for VZV gE and a comprehensive analysis of the epitope landscape on gE. The limited number of Abs used in this study were found to predominately recognize the gI binding domain of gE. It will be interesting to see if this binding pattern continues in future studies as well as for Abs elicited through vaccination with RZV. Currently, licensed vaccines are not available for HSV-1 and HSV-2. The gE constructs used in this work, particularly gE_gIBD, may serve as useful templates for studying the structure of gE from these herpesviruses, which may help accelerate vaccine design and development efforts. 

## Figures and Tables

**Figure 1 vaccines-12-01111-f001:**
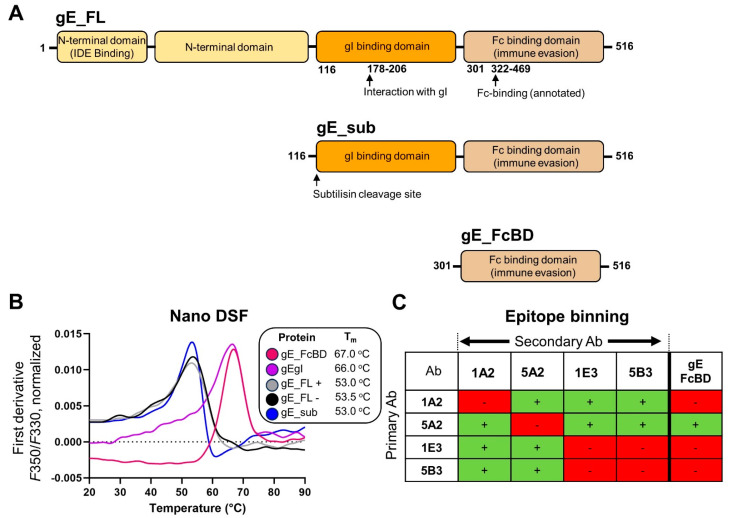
Characterization of VZV gE constructs and anti-gE Fabs. (**A**) Schematic of VZV gE illustrating the annotated domains, and the location for indicated residues. (**B**) Nano-DSF analysis of the melting points of gE constructs. The gE_FcBD and gEgI heterodimer are each more thermostable than gE_sub or fully glycosylated (+) and deglycosylated (−) gE_FL. (**C**) Results of an HPLC-based Ab binning assay and binding to gE_FcBD. In the columns between the dotted lines, the red boxes displaying an “−” indicate that the primary and secondary Abs compete with each other, whereas a green box displaying “+” indicates no competition. The final column in the table indicates whether a primary Ab on the Y-axis binds to the gE_FcBD construct.

**Figure 2 vaccines-12-01111-f002:**
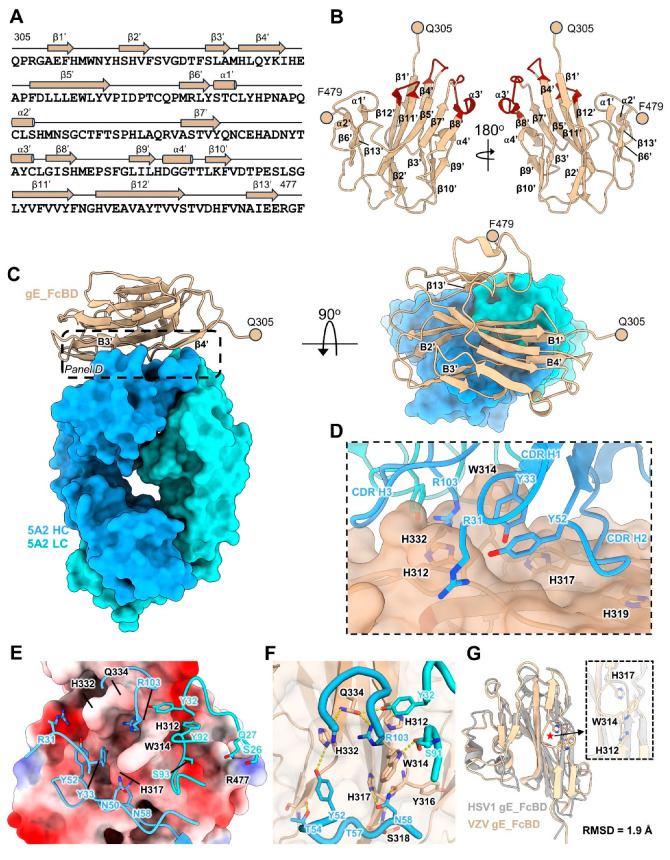
Structure of the gE_FcBD in complex with Fab 5A2. (**A**) Sequence for the visible portion in the X-ray structure of the gE_FcBD. Residues involved in β-strands and α-helices are indicated with brown arrows or tubes, respectively. (**B**) Two views of the structure of the gE_FcBD shown in cartoon representation. The “CDR” loops are colored red. (**C**) Two views of the gE_FcBD:5A2 Fab complex. The view on the right is rotated by 90° towards the plane of the reader. The 5A2 Fab is shown in surface representation with the heavy and light chains colored blue and cyan, respectively. (**D**) Zoomed in view of the epitope/paratope interface from panel (**C**). Fab 5A2 is shown in cartoon and gE_FcBD is shown in transparent surface and cartoon. The cluster of histidine on the epitope and 5A2 residues that bury surface on them are shown as sticks. (**E**) The surface of gE_FcBD is colored based on electrostatic potential: red = negative; blue = positive; white = neutral/hydrophobic. (**F**) Hydrogen bond interactions between 5A2 and gE_FcBD. (**G**) Superposition with the HSV gE X-ray structure showing the high similarity. The zoom window shows potential plasticity in the 5A2 epitope for the loop between His312 and His317.

**Figure 3 vaccines-12-01111-f003:**
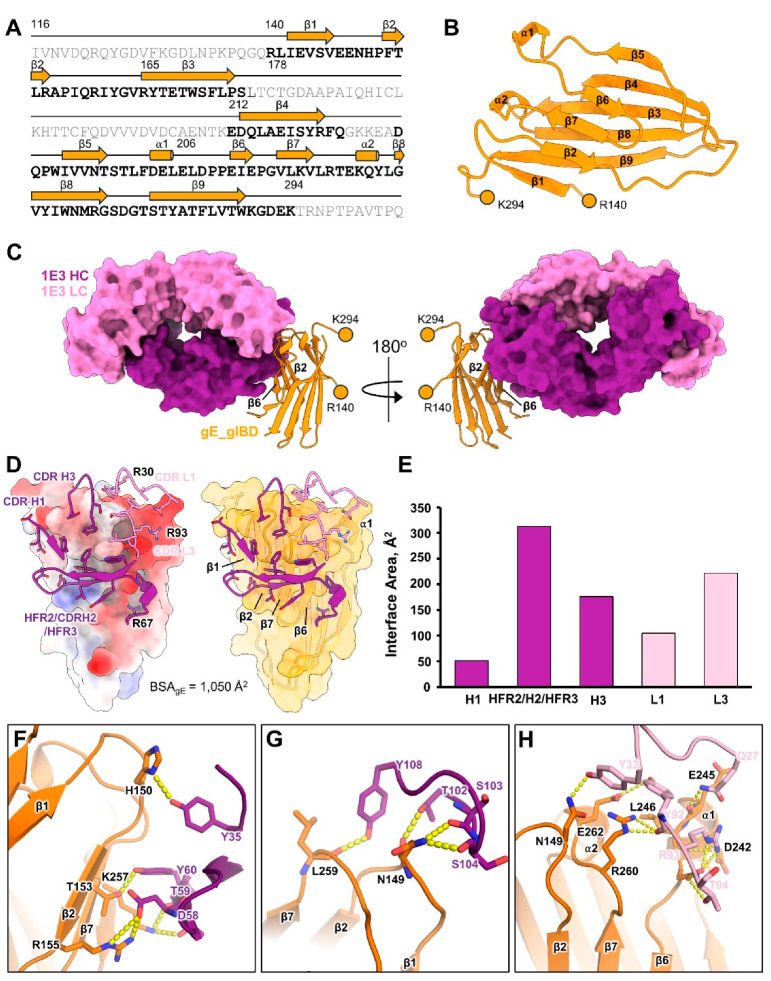
Structure of gE_gIBD in complex with Fab 1E3. (**A**) Sequence for the gE_gIBD construct. Residues that are visible in the electron density maps are colored black. Residues involved in β-strands and α-helices are indicated with orange arrows or tubes, respectively. (**B**) X-ray structure of the gE_gIBD shown in cartoon representation with β-strands and α-helices labeled. The N- and C-terminus residues are also indicated. (**C**) Two views of the gE_gIBD structure shown in complex with Fab 1E3, which is displayed in surface representation. (**D**) The gE_FcBD is shown with surface charge potential (**left**) or with transparent surface and cartoon (**right**). The 1E3 heavy- and light-chain CDRs and heavy-chain FRs that contact gE are shown in cartoon, with select residues shown as sticks to illustrate shape and charge complementarity. (**E**) Graph showing BSA for 1E3 CDRs. (**F**) Hydrogen bond interactions between gE and the 1E3 heavy chain CDRs 1 and 2 or (**G**) CDR H3. (**H**) Hydrogen bonds between gE and the 1E3 light chain.

**Figure 4 vaccines-12-01111-f004:**
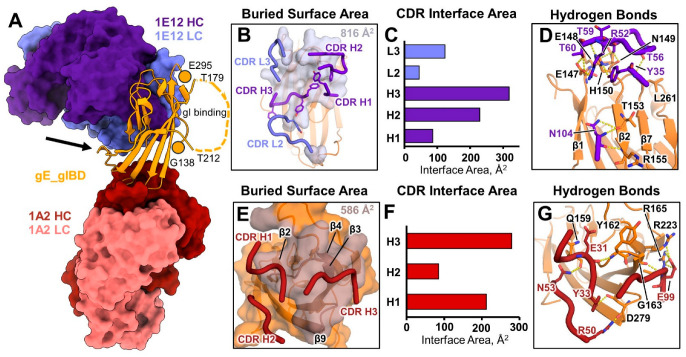
Epitope mapping of the gE_gIBD. (**A**) X-ray structure of the gE_gIBD in ternary complex with Fabs 1A2 and 1E12, which are shown in surface and colored as labeled in the panel. The black arrow indicates a loop on gE_gIBD, which was not resolved in the 1A2-bound structure. The residues forming the gIBD are depicted with a dashed yellow line. (**B**) The 1E12 CDRs that contact gE are shown in cartoon tubes and sticks and the BSA on gE is shown in transparent surface. (**C**) Graph indicating the 1E12 CDR interface areas on gE. (**D**) Hydrogen bond interactions between gE and Fab 1E12. (**E**) The 1A2 CDRs that bury surface on gE are shown as cartoon tubes and gE is shown in transparent surface with the regions buried by 1A2 shaded dark. (**F**) Graph indicating the 1A2 CDR interface areas on gE. (**G**) Hydrogen bond interactions between gE and Fab 1A2.

**Figure 5 vaccines-12-01111-f005:**
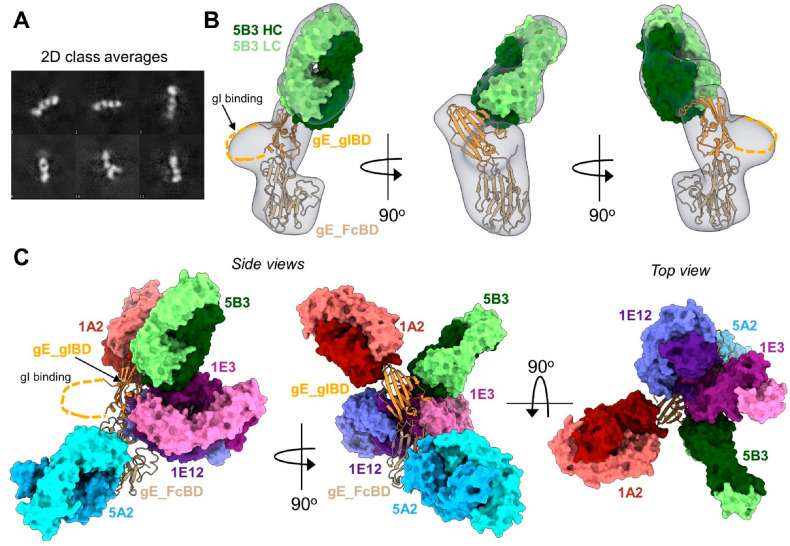
Cryo-EM analysis of gE_FL and epitope mapping of Fab 5B3. (**A**) Cryo-EM 2D classes for FL gE in complex with Fab 5B3. (**B**) Low-resolution 3D reconstruction of the FL gE:5B3 Fab complex. The gE_gIBD and gE_FcBD X-ray structures, and a 5B3 homology model were fitted in the EM density. Unfit density is presumed to be the gIBD and is indicated with a dotted orange line. The disordered N-terminal domain was not visible. (**C**) The fitted structures from (**B**) docked with the 5A2, 1E3, 1E12, and 1A2 Fabs to their respective epitopes. The middle view best displays how one face of gE is not recognized by the Abs used in this study.

## Data Availability

All data needed to evaluate the conclusions in the paper are present in the paper and/or the Appendix A. The atomic coordinates and corresponding structure factors have been deposited in the RCSB Protein Data Bank under the following PDB codes: gE_FcBD:5A2 Fab (PDB ID 8V5P); gE_FcBD:5A2 Fab (cleaved) (PDB ID 8V5S); gE_gIBD:1E3 Fab (PDB ID 8V5Q); gE_gIBD:1A2:1E12 Fab ternary complex (PDB ID 8V5L).

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
