# Peer review of "Structures of the Varicella Zoster Virus Glycoprotein E and Epitope Mapping of Vaccine-Elicited Antibodies"

_vaccines, 2024, doi:10.3390/vaccines12101111_

Round 1

Reviewer 1 Report

Comments and Suggestions for Authors

The manuscript analyzed the antibody epitopes on the gE protein involved in examining the complex formed between antibodies produced by the immune population and the gE protein. While this study holds notable significance, several issues warrant further clarification. The specific recommendations are as follows:

1.  Why was the antibody produced after immunization with Zostavax, a live attenuated vaccine, chosen for epitope analysis instead of antibodies from the Shingrix-immunized population?

2. Whether previous studyspecify the targets of the four antibodies? It is advisable to include a description of the characteristics and specific targets of these antibodies.

3.  What role does the N-terminal domain of the gE protein play, and are there antibodies that target this region? This information should be clarified.

4. Figure 1 presents the three proteins gEgI, gE FL+, and gE_FL-. It is recommended to include schematic diagrams in Figure A to better illustrate these proteins.

5. In line 324, the term “antibody sequence” might be more accurately described as “antibody.” Please review and revise as necessary.

6. How can the structural analysis of the four antibodies contribute to vaccine design? It would be beneficial to discuss potential implications and future prospects in this regard at the end of the manuscript.

Reviewer 2 Report

Comments and Suggestions for Authors The work in the manuscript describes the complex crystallographic analysis of the gE Varicela zooster virus and its domains, using specific recombinant antibodies. The wide crystal structure obtained in the manuscript and molecular simulation further elucidate the immunogenicity of the epitopes also used for vaccination. Advanced biochemical and analytical procedures for the molecular structure of the glycoprotein study described are of high level, such as analytical pretreatment and epitope mapping techniques. The authors documents the discussion and conclusions by large amounts of illustrative figures and tables, also in supportive material. This topic is currently studied by different groups and in my opinion, this work shifts the topic to better understand the mechanism or efficiency of this vaccine.

Shortcuts (i.e. gE_FL, FcBD etc.) might be explained sooner in the Introduction, while it is used further in the result and other parts.

Reviewer 3 Report

Comments and Suggestions for Authors

The manuscript “Structure of the varicella zoster virus glycoprotein E and epitope mapping of vaccine elicited antibodies” shows the molecular structure of VZV gE, an IgG-like fold composed of nine beta-strands, in complex with six different human neutralizing Fabs previously obtained after the ZVL vaccination, to characterize the humoral immune response to this antigen. The current version of the manuscript is well written, the results are described in excellent way.  Only I have minor comments:

Minor comments

-              Include an alignment where the interacting residues recognized by all the antibodies are shown.

-              Discuss more about the relation between the epitopes recognized by the Fab and their biological activity (in vitro IC50 values), previously obtained.

-              Discuss more about the presence of convergent residues of the gIBD epitopes recognized by the human antibodies and their importance. 
